# Methods of Analyses for Biodegradable Polymers: A Review

**DOI:** 10.3390/polym14224928

**Published:** 2022-11-15

**Authors:** Siti Baidurah

**Affiliations:** 1Bioprocess Technology Division, School of Industrial Technology, Universiti Sains Malaysia, Minden 11800, Malaysia; sitibaidurah@usm.my; Tel.: +60-4653-6381; 2Green Biopolymer, Coatings & Packaging Cluster, School of Industrial Technology, Universiti Sains Malaysia, Minden 11800, Malaysia

**Keywords:** biodegradation, biodegradable polymers, polyhydroxyalkanoates, chromatographic

## Abstract

Biodegradable polymers are materials that can decompose through the action of various environmental microorganisms, such as bacteria and fungi, to form water and carbon dioxide. The biodegradability characteristics have led to a growing demand for the accurate and precise determination of the degraded polymer composition. With the advancements in analytical product development, various analytical methods are available and touted as practical and preferable methods of bioanalytical techniques, which enable the understanding of the complex composition of biopolymers such as polyhydroxyalkanoates and poly(lactic acid). The former part of this review discusses the definition and examples of biopolymers, followed by the theory and instrumentation of analytical methods applicable to the analysis of biopolymers, such as physical methods (SEM, TEM, weighing analytical balance, etc.), chromatographic methods (GC, THM-GC, SEC/GPC), spectroscopic methods (NMR, FTIR, XRD, XRF), respirometric methods, thermal methods (DSC, DTA, TGA), and meta-analysis. Special focus is given to the chromatographic methods, because this is the routine method of polymer analysis. The aim of this review is to focus on the recent developments in the field of biopolymer analysis and instrument application to analyse the various types of biopolymers.

## 1. Introduction

The plethora of current knowledge of biodegradable polymers has been supported by the selection of various analytical methods used by researchers. Since polymer degradation is an intricate process impacted by various parameters, it is improbable that any one selected method could provide a comprehensive picture of the changes in polymer degrading properties at both the macroscopic and chemical structure levels. Cross-comparisons of two or more independent methodologies can provide deeper knowledge of the degradation characteristics. The preferred approach for quantitatively measuring biodegradation rates should be simple, precise, quick, and economical. Comparisons of information collected using several analytical techniques should be conducted with caution, taking into account many parameters (the number of samples required, degree of sensitivity, etc.) that may contribute to inappropriate comparisons and inaccurate judgments. This review discusses the broad concepts, benefits, and drawbacks of the many methodologies available for studying biodegradable polymers, such as physical observation, chromatographic, spectroscopic, and respirometric methods, and meta-analysis, as delineated in Figure 1. In addition, the gaps in, and development of, the analytical methods are also highlighted.

Biodegradable polymers are materials that can be decomposed through the action of various environmental microorganisms, such as bacteria and fungi, to form water and carbon dioxide [1]. The biodegradation mechanisms or decomposition begins on the polymer surface due to the action of the extracellular enzymes of microorganisms, generating oligomers. These corresponding oligomers then enter the microorganism cell, in which they act as carbon sources and are metabolized into carbon dioxide and water [1]. Biopolymers have garnered a great deal of interest as “green” or “environmentally friendly” polymetric materials owing to their degradability properties and low environmental load upon disposal [2,3,4,5,6,7]. To enhance their physical and thermo-chemical properties, biopolymers are often enhanced to improve their suitability for their final product applications. The enhancement is achieved by the incorporation of fillers, binders, or copolymers. These modified biopolymers are applied and produced widely on the industrial scale [8].

### 1.1. Examples of Biopolymers

Table 1 summarizes the classification of the current commercially available biodegradable polymers based on their origins, together with the trade names and manufacturers [9,10]. These polymers generally consist of polyesters and polysaccharides bearing hydrolysable ester or ether bonds in their backbones, respectively. They can be categorized according to their origins of production, namely bacteria, natural products, and chemical synthesis.

#### 1.1.1. Biodegradable Polymers Produced by Bacteria

Some polyesters and polysaccharides accumulate in bacteria as a source of intracellular energy and carbon [9,11,12,13,14,15,16,17,18,19,20,21]. Polyhydroxyalkanoates (PHAs) are biodegradable aliphatic polyesters formed entirely through bacterial fermentation. *Alcaligenes eutrophus* has been extensively studied due to its capacity to generate vast volumes of poly(3-hydroxybutyrate) (PHB). PHB accumulation in *A. eutrophus* can be controlled by changing the types or concentrations of carbon and nitrogen sources. For example, *A. eutrophus* produced PHB at more than 80% of the dry weight upon culture in a medium with an abundance of carbon sources, such as glucose, and low amounts of nitrogen sources.

In general, the brittleness of PHB, which is caused by its high crystallinity, has limited its application on the industrial scale. Various copolymers, such as 3-hydroxyvalerate (3HV) units, are frequently incorporated into the polymer chain through bacterial fermentation to increase its toughness and flexibility. Because of its great biocompatibility and non-toxicity, the resultant copolymer of poly(3-hydroxybutyrate-*co*-3-hydroxyvalerate) (P(HB-*co*-HV)) is employed to create internal sutures in the biomedical industry [8,22].

Gellan gum and curdlan are biodegradable polysaccharides produced by bacteria. They have mainly been utilized as food additives, especially gelling and thickening agents, owing to their high water absorbency and nontoxicity [9,23]. Edible films using gellan gum and curdlan have recently been developed and used as for food wrapping owing to their high water vapor permeability [24].

#### 1.1.2. Biodegradable Polymers Produced from Natural Products and Their Derivatives

Despite the fact that chitin is one of the most prevalent natural compounds after cellulose, its low solubility and reactivity have restricted its industrial and commercial applicability. To solve this problem, chitin has been chemically modified by grafting with synthetic polymers to improve its miscibility with various commodity polymer [9]. Aoi et al. synthesized chitin derivatives containing polyoxazoline side chains and prepared miscible blends containing synthetic polymers, such as polyvinyl chloride and polyvinyl alcohol. These blends are widely used as new polymetric materials not only for their biodegradability but also for their moulding and mechanical properties, which are akin to those of commodity polymers [25].

In addition, PHB has also been produced using the leaves of transgenic plants such as *Arabidopsis thaliana*. This plant-mediated synthesis was justified by its potential production on a greater scale [26] at a lesser cost than microbial fermentation. As reported by Proirer et al., only the first enzyme necessary for PHB production from acetyl-CoA, 3-khetotiolase, is endogenously present in plants. *A. eutrophus* genes encoding acetoacetyl-CoA reductase and PHA synthase were expressed in transgenic *A. thaliana* to accomplish the PHB synthesis pathway in plants [27]. Bohmert-Tatarev et al. patented the methods of stable, fertile, and high PHA production in plants [28].

#### 1.1.3. Biodegradable Polymers Produced via Chemical Synthesis

This group mainly consists of polylactic acid (PLA), polyglycolic acid (PGA), polycaprolactone (PCL), poly(butylene succinate-*co*-butylene adipate) (PBSA), and other aliphatic polyesters. These polymers are synthesized commonly through ring-opening polymerization accelerated by metals (ROP) or the polycondensation of their corresponding petroleum-derived monomers [9,29]. Among these polymers, PLA, which can be prepared from natural products, such as cereal- or sugarcane-based saccharides, as well as petroleum precursors, has attracted tremendous attention as a biopolymer with application potential [30,31]. Lactic acid that is obtained from glucose or sucrose via *lactobacillus* fermentation is polymerized by ROP after dimerization or direct polycondensation [8]. PLA typically shows a high rigidity, making it a suitable replacement for polystyrene and polyethylene terephthalate (PET) in several applications, such as packaging and textiles [30].

The biodegradability of commodity synthetic polymers can be increased by blending them with several types of natural products, such as starch and chitin. Ratto et al. prepared films from a poly(butylene succinate/adipate) (PBSA)/starch composite and investigated the processability and biodegradability of the resulting films, along with their mechanical and thermal properties. The biodegradable PBSA/starch film exhibited sufficient mechanical properties for plastic extrusion applications [32].

Similarly, PBS, which is created by polycondensing 1,4-butandiol and succinic acid, has found usage in a broad range of applications because its physical qualities are similar to the properties of commercially available polymers, such as polypropylene (PP) and polyethylene (PE). Additionally, monomers such as butylene adipate units are typically added into PBS polymer chains to enhance their strength and elasticity. The resulting copolyester, viz., PBSA, has also found several uses, such as agricultural and construction materials [32].

As mentioned above, various types of biodegradable polymers are utilized in numerous fields ranging from agricultural to biomedical applications. As citizens become more conscious of the importance of waste disposal and ecological preservation, biodegradable polymer usage is expected to increase and eventually replace the current commodity polymetric materials in practical use. The synthesis and/or modification of biodegradable polymers with improved mechanical and physical features have also piqued the attention of researchers. In the foreseeable future, the effective production of polymers exhibiting improved properties at reduced costs may become more crucial for realizing a sustainable society. Various initiatives have been proposed to lower the manufacturing costs of biopolymers, among which is a method based on utilizing industrial by-products such as molasses, waste glycerol, and banana frond extract [33,34,35,36,37,38,39].

### 1.2. Biodegradation Conditions

It is vital to understand the mechanisms and kinetics of polymer degradation in various natural and controlled settings in order to predict polymer degradation behaviour in dynamic environments [40]. Researchers are enthusiastically engaging in the biodegradation research of polymers in a variety of natural and controlled settings. Soil, mangrove wetlands, seas, rivers, anaerobic sludge, activated sludge, and aerobic or anaerobic compost are examples of natural settings [41,42,43,44]. The biodegradation test is also performed in vivo and in vitro to investigate the material’s biocompatibility property, for example, in phosphate buffer, blood serum, human blood, and animal muscle tissue [45,46]. After the complete biodegradation of polymers under aerobic conditions, the end products are carbon dioxide and water, whilst methane is the end product under anaerobic conditions [41].

Several variables influence polymer biodegradability, viz., climate, soil/water characteristics (humidity, temperature, pH, oxygen and nutrient levels), the indigenous microorganism consortia in the setting, and physiochemical qualities (surface area, chemical composition, polydispersity index, molecular weight, crystallinity) [41,47,48,49,50,51,52,53,54,55,56,57,58].

### 1.3. Importance of Biopolymer Analyses

The compositional data and remaining products of biopolymers often provide useful clues for the prediction of the materials’ biodegradability and physical properties. These benefits have led to a growing demand for the accurate and precise determination of the biodegradable polymer composition. Both qualitative and quantitative analyses of biodegraded materials are possible. Among these approaches are physical methods, respirometric, chromatographic, and spectroscopic methods, and meta-analysis. Among these methods, chromatographic methods are the most frequently utilized characterization techniques. Their applications and limitations are discussed below.

## 2. Physical Methods

Physical methods entail the physical observation of polymers and their surface micro-morphology, strength qualities, and weight reduction. Table 2 delineates the summarization of numerous methods to assess the biodegradation of polymers in various experimental settings, together with their advantages and limitations.

### 2.1. Physical Observations

Electron microscopy, such as scan electron microscopy (SEM), enables the observation of surface deterioration [59], whilst transmission electron microscopy (TEM) is widely used to observe the ultrathin cross-section of polymeric samples [60]. For example, PHA biodegradation, or bioerosion, is triggered on the polymer’s surface by PHA depolymerase. This qualitative examination enables the visible physical observation of the polymer’s hole, coarse, and porosity texture, which encourages bacterial attachment and leads to PHA depolymerase secretion [61,62]. The large size of the porous surface area occurs as a result of the microorganism’s reaction and, therefore, can be interpreted as the degree of biodegradation.

An experimental investigation was conducted by Adamcova et al. to visually monitor the microstructure and morphology of seven commercial packaging bioplastics (starch, polycaprolactone, natural material, cellulose, some material compositions that are not stated, etc.) and one petrochemical plastic claimed to be biodegradable (high-density polyethylene with totally degradable plastic additives, HDPE + TDPA) throughout a soil compost test for 12 weeks. The three samples with additives, including HDPE + TDPA, and samples with unknown material compositions did not indicate any visual signs of degradation and no colour changes in contrast to four samples that portrayed significant erosion, breaches, fractures, and holes on the surface when analysed by SEM [61].

### 2.2. Strength Properties

A high-precision tool called an extensometer is specially created to elucidate the strength properties of polymeric materials, such as the tensile strength, yield strength, yield point elongation, and strain ratio. Tensile strength can be defined as the highest stress that a polymeric plastic material can sustain while being stretched before breaking [63]. The tensile strength will decrease as the biodegradation process progresses and can be measured using an extensometer [64].

An extensometer is not only a tool for the elucidation of polymer deterioration/degradation but can also be applied to monitor the viscoelasticity of newly ameliorated polymers. Morreale et al. employed a commercial biopolymer produced by FKuR Kunststoff GmbH (Willich, Germany), with the trade name of BioFlex F2110, which is based on a blend of PLA and thermoplastic-copolyesters, and then the authors modified it by adding wood flour. These composites were then subjected to a viscoelasticity test at 60 °C using an oven outfitted with four extensometers directly attached to movable clamps and weight holders of 1.5 MPa. The acquired results were plotted against time and demonstrated that the addition of wood flour improved the rigidity and viscoelasticity resistance by 0.6% when compared to the pristine polymer, without compromising other parameters, such as the tensile strength [65]. 

**Table 2 polymers-14-04928-t002:** Summarization of numerous methods to assess the biodegradation of polymers in various experimental settings, together with their advantages and limitations.

No.	Test Material	Brief Description, Aim, and Degradation Condition	Advantages or Limitation	Ref.
Physical methods				
1	PHBHHx/PBAT (80/20)	TEM: To analyse the cross-section of the sheet. SEM: To analyse the surface morphology of the melt blend sheet. Ruthenium tetroxide (RuO_4_) is used as a staining agent. Condition: Seawater for 28 days with 42% biodegradation.	TEM: The PBAT region darkens owing to the lower permeability of the electron beam through the attachment of RuO_4_ to the phenyl group in the PBAT as compared with the PHBHHx region, allowing for clear observation.	[60]
2	PHB, P(HB-*co*-12%-HHx)	SEM: To analyse the surface morphology. Conditions: Activated sludge for 18 days with a 0.65% recovery weight.	SEM: Able to observe the porous and rough surface, indicating fast degradation, also contributing to the low degree of crystallinity of HHx.	[62]
Chromatographic methods				
3	P(HB-*co*-HHx) films.	Thermally assisted hydrolysis and methylation–gas chromatography (THM-GC): Rapid analysis of the copolymer composition during biodegradation. Condition: Farm soil pH 5.3 for 28 days at 34 °C with 1.91% recovery weight.	THM-GC: Able to observe the local biodegradation behaviour of the degraded films based on the changes in the copolymer composition.	[44]
4	Intracellular P(HB-*co*-HV) in *Ralstonia eutropha* and recombinant *Escherichia coli.*	Py-GC-MS: Rapid analysis of PHA contents and their monomer compositions accumulated intracellularly	Py-GC-MS: Promising tool used to rapidly screen PHA-containing strains based on polymer contents, along with their monomer compositions. The data obtained by this method indicate results similar to those of conventional GC-FID.	[66]
5	Whole bacterial cells of *Cupriavidus necator* accumulating P(HB-*co*-HV).	THM-GC in the presence of TMAH: Rapid and direct compositional analysis of P(HB-*co*-HV) in whole cells.	THM-GC in the presence of TMAH: Chromatograms clearly indicate peaks derived from the HB and HV units of the polymer chains without any interference by the bacterial matrix components and no cumbersome sample pre-treatment required. The data obtained by this method indicate results similar to those of conventional GC-FID.	[67]
Spectroscopic methods				
6	Extracted PHAs from bacteria cells such as *E. coli* and *Pseudomonas* sp.	NMR: To characterize PHAs. FTIR: To investigate functional groups of the PHAs.	FTIR data indicate the presence of hydroxyoctaoate, medium-chain-length PHAs and hydroxydecanoate with strong bands at 1631 cm^−1^, 1548 cm^−1^, and 1409 cm^−1^. NMR spectra show the presence of interconnection functional groups of HC=CH bonds at 3.363 ppm and CH_2_O-COOH bonds at 2.548 ppm. Both methods require a sample extraction process prior to each analysis.	[68]
7	Bioplastics from starch/chitosan reinforced with PP.	FTIR: To investigate functional groups of the bioplastics. The spectra portray main bonds of O-H hydrogen bonds (carboxylic acid), C-H alkanes, C=C alkenes, and C-O alcohols.	FTIR spectrum indicates that the functional groups of bioplastics have similarity with their constituent components. Sample extraction process is required.	[69]
8	Bioplastics from starch/chitosan reinforced with PP.	XRD: To analyse the degree of crystallinity.	The XRD data evince that the biopolymer had an amorphous crystalline structure, with the major wide peaks located between 18° and 30°.	[69]
9	Single-use bioplastics (SUPs).	XRF: To determine the elemental composition and concentration in the SUPs.	Most of the SUPs exceed the standard values, and the highest concentrations of Cu, Cr, Mo, Zn, Fe, and Pb were 1898 mg/kg, 1586 mg/kg, 95 mg/kg, 1492 mg/kg, 1900 mg/kg, and 7528 mg/kg. XRF is a non-destructive analytical method which enables the same sample to be used again for further analysis.	[70]
Respirometric methods				
10	PHBHHx/PBAT (80/20).	BOD tester: To measure biodegradation induced by the microbial metabolism. Condition: Seawater for 28 days with 42% biodegradation.	The seawater was placed together with samples in a fermenter, and the amount of consumed oxygen (O_2_) gas was measured. This method is an indirect measurement of O_2_ that has been utilized for complete sample degradation, which may carry some errors. It is preferable to cross-check the results with the weight loss data.	[60]
Meta-analysis				
11	PHAs	Statistical analysis: To estimate the PHA mean biodegradation rate and lifetime. Condition: Seawater	The statistical analysis enables the estimation of the biodegradation of PHA in the seawater as 0.04–0.09 mg/day/cm.	[71]
Thermal methods				
12	Poly(butylene succinate-*co*-butylene adipate) film sample	DSC: To determine the degree of crystallinity.	The obtained degree of crystallinity of the original and heated film samples were 46.1 and 42.4%, respectively, evinced the crystallinity of the PBSA film, which was considerably lower for the heat-treated film.	[48]
13	PHAs accumulated in *Pseudomonas putida*	DSC and TGA: To analyse the thermal properties.	Thermal properties indicate that the accumulated PHA is semi-crystalline, with a good thermal stability, *T_d_* of 264.6 to 318.8 °C, *T_m_* of 43 °C, *T_g_* of −1.0 °C, and Δ*H_f_* of 100.9 J/g. Requires sample pre-treatment of PHA extraction.	[72]

### 2.3. Mass Reduction

Measuring the weight loss is the simplest practical, direct, and widely used method of quantifying the biodegradation activity of any polymeric material using an analytical balance. Baidurah et al. conducted a soil burial degradation test within 28 days to elucidate the degradation mechanisms of poly(butylene succinate-*co*-butylene adipate) (PBSA) thin film samples with a ratio of 82.2:17.8. After each designated period of 7, 14, 21, and 28 days, the deteriorated PBSA films were cleaned, dried, and weighed. The recovery (weight %) of each deteriorated film was obtained by dividing its dry weight by the weight of the original film prior to the burial test. Upon 28 days of the soil burial degradation test, the recovery weight of the material was reduced to 68.1% [73]. Researchers should bear in mind that, when using this method, the mass reduction should be calculated by deducting the weight of the original sample prior to the degradation test.

Due to the fact that weight loss measurements are difficult to extrapolate, it is preferable to complement this approach with other analytical methods, such as respirometric methods, FTIR, and NMR [49]. In a separate study by Salomez et al., the authors conducted both experiments of the weight loss and respirometric methods simultaneously to compare the degradation of two samples, viz., P(HB-*co*-HV) and PBSA. In the respirometric analysis, the carbon dioxide released by the materials during their degradation were recorded. Upon 450 days of incubation, they reported that the P(HB-*co*-HV) and PBSA polymers had degraded, with weight losses of 5.5 and 8.0%, respectively. The authors emphasized that the polymer’s weight loss does not ensure its eventual absorption by microorganisms, as demonstrated by respirometry analysis [74].

## 3. Chromatographic Methods

Chromatographic methods are described as a process whereby a chemical mixture transported by a gas or liquid (mobile phase) is separated into components as a consequence of the differential distribution of the solutes as they flow around or over a stationary solid or liquid phase (stationary phase). This approach is commonly used to separate, identify, and determine the chemical components of complicated mixtures and polymers. In comparison to other methods, the chromatographic methods are touted as a powerful and practical tool for the separation and identification of polymers.

### 3.1. Gas Chromatography

The chemical composition of biopolymers can be analysed using gas chromatography (GC) after sample pre-treatments, such as transmethylation and solvent extraction. Methylation improves the determination of samples such as polymeric materials and fatty acid components in essential oils. Methylated samples produce clear chromatographic peak shapes, without any undesirable fronting or tailing peaks. Thus far, this technique has been used widely used to elucidate the chemical composition of numerous biodegradable polymers, such as PHB [66] and P(HB-*co*-HV) [44,66,67,75,76]. Furthermore, this method can quantitatively analyse the amount of biodegradable polymer accumulated in bacteria [66,75,76]. The sample pre-treatment procedure developed by Braunegg et al. in the 1970s [77] remains useful for polyester characterization in bacteria. In a study using this method, centrifuged bacterial cells were suspended in an acidic methanol solution (ca. 10% H_2_SO_4_) and chloroform. The mixture was then heated to 100 °C for a few hours to depolymerize and transmethylate the polyesters present in the cells into their constituent monomers. After cooling to room temperature, the transmethylated monomers were separated utilising water as a medium through liquid–liquid extraction and subsequently analysed by GC [77].

However, this GC method requires a substantial sample quantity (at least 20 mg) and a fairly long pre-treatment duration (about half a day) prior to the final GC analysis. This is considered a drawback, because of which it is difficult to obtain approximately 20 mg of the degraded polymer. Therefore, to solve this problem, a practical and highly sensitive approach is required.

### 3.2. Thermally Assisted Hydrolysis and Methylation–Gas Chromatography

Thermally assisted hydrolysis and methylation–gas chromatography (THM-GC) can measure changes in the chemical composition ratio of polymeric materials. This method is also known as reactive pyrolysis–gas chromatography (Reactive Py-GC) [78,79]. THM-GC, in the presence of organic alkali, allows for the analysis of deteriorated film samples at a particular local spot or point [48,73]. Furthermore, this approach effectively achieved the high-sensitivity, direct, and fast measurement of intracellular PHAs without any laborious and time-consuming sample preparations, such as solvent extraction [80,81,82].

The instrumentation of THM-GC consists of a microfurnace pyrolyzer (Frontier Laboratories Ltd., Fukushima, Japan) attached to a GC equipped with a flame ionization detector (FID). In most studies, a minute amount of the sample in liquid or solid form (approximately 20 µg) was placed in a small platinum cup, and 2 µL of tetramethylammonium hydroxide (TMAH) solution as a methylating reagent was added. The sample cup was initially placed at room temperature in the pyrolyzer’s waiting position and then dropped into the heated core of the pyrolyzer, which was maintained at 350–500 °C by the flow of helium carrier gas at a rate of 50 mL/min. A portion of the flow (1 mL/min) was reduced by a splitter and injected into a metal capillary separation column coated with immobilised 5% diphenyl-95% dimethylpolysiloxane (1.0 m film thickness) (Frontier Laboratories, Ultra ALLOY-5 (MS/HT); 30 m long × 0.23 mm i.d.). The temperature of the column was set from 50 to 300 °C at a rate of 5 °C/min. A GC–mass spectrometry (GC-MS) system (QP-5050 Shimadzu Corporation, Kyoto, Japan) with an electron ionisation (EI) source was used for the peak identification [48].

In 1989, Challinor reported that the pyrolysis–gas chromatography (Py-GC) method with organic alkalis, such as TMAH, is effective for the analysis of synthetic polymers and natural organic compounds containing ester bonds and/or polar groups [83]. During thermochemolysis, in the presence of TMAH, only ester bonds or ether bonds in the sample components are hydrolysed and subsequently methylated to yield their corresponding methyl derivatives, while carbon–carbon bonds are maintained. Based on the extremely simplified thermograms consisting mainly of the methyl derivatives of the sample constituents, the qualitative and quantitative analysis can be performed without any cumbersome sample pre-treatment. This report by Challinor greatly progressed the field of analytical pyrolysis, and this technique has since been utilized to successful characterize condensation polymers [48,83,84] and natural polar organic compounds, such as lipids [85,86], natural resins [87,88], natural waxes, humic substances [89], and lignin [90,91].

TMAH, which is the most widely used organic alkali in THM-GC, has several advantages [92]: (1) TMAH can react with samples not only as a hydrolysis reagent but also as a methylating reagent, with a high reaction efficiency, and (2) TMAH itself is decomposed into tetramethylamine and methanol at temperatures above 130 °C, which do not damage the separation column. Various terms have been employed to describe the reactive pyrolysis of ester compounds in the presence of TMAH via the above-described mechanism, including thermochemolysis and on-line methylation [48].

To date, numerous condensation polymers, including multi-component liquid crystalline aromatic polyesters (LCPs), cationic polyacrylamide resins, copolymer type polycarbonates (PCs) [93], poly(aryl ether sulfone) [94], and P(HB-*co*-HV) [95,96,97], have been subjected to compositional analysis using THM-GC in the presence of TMAH. Among these papers, Sato et al. used the THM-GC method for the compositional analysis of a biodegradable copolyester, P(HB-*co*-HV), with various ratios of the 3HV contents (ca. 4–23 mol%). The report by Sato et al. showed that P(HB-*co*-HV), after pyrolysis in the presence of TMAH at 350 °C, underwent not only standard reactive pyrolysis but also the *cis*-elimination of the ester linkages, followed by the hydrolysis and methylation of the nearby ester bonds. Based on the peak intensities of the products generated by these reactions, the copolymer composition was precisely analysed, without the use of any cumbersome pre-treatment procedure. Furthermore, the obtained copolymer compositions were in agreement with those determined by ^1^H-NMR [95].

Research conducted by Baidurah et al. showed that THM-GC was able to rapidly and accurately detect the composition of PHB produced in *Cupriavidus necator*. Only trace amounts of the samples as low as 100 ± 5 µg of the dried cells were required in THM-GC to provide a precise analysis of the PHB composition. The data obtained by THM-GC and the conventional GC analysis method were compared. Although there was a slight discrepancy observed via the THM-GC method, the general trend of the analysis showed a linear correlation for both methods, with r^2^ = 0.9972. The slight discrepancy was caused by the additional detection of the presence of PHB units in both the polymer chains and monomer compounds. The results evinced THM-GC can be effectively and practically applied to the composition analysis of PHAs produced by microbial cells, without any tedious sample pre-treatment [81].

In another study, Khok et al. elaborated on the comparison of quantification methods between conventional GC and THM-GC to characterize the PHB content [82]. The authors analysed the PHB content accumulated in *Bacillus* sp. when cultured in various molasses concentrations, ranging from 10 to 20 and 30 mg/L, by THM-GC and compared the data with the conventional GC method. The values of the PHB contents corresponded, overall, with those obtained by the conventional GC method, with a high correlation coefficient (r^2^) of 0.9766.

### 3.3. Hyphenated Method of GC-MS

When two or more analytical techniques or instruments are combined to form a new, more efficient device, the resulting methodology is often termed as a hyphenated or integrated method. GC is an ideal way of introducing mixtures, because the components are separated from the mixture by the chromatograph prior to their introduction to the mass spectrometer (MS). The combination of GC and MS is called GC-MS. The separation process takes place in the GC, followed by the identification of the separated homopolymers by the MS. For the study of complex biochemical mixtures, as well as biodegradable polymers, GC-MS has emerged as one of the most potent technologies available at present. For example, in biodegradable polymer application, the obtained spectra are created from compounds as they exit a chromatographic column. Then, these spectra are stored by a computer for the subsequent process of peak identification in MS. The MS can also be coupled with liquid chromatography (LC-MS) for the analysis of samples comprised of non-volatile constituents.

Khang et al. [66] reported the characterization of PHA contents and the monomer composition of microbial cells based on pyrolysis–gas chromatography combined with mass spectrometry (Py-GC-MS). It is well-known that various kinds of microbial cells are able to accumulate different PHA contents and monomer compositions, such as *Ralstonia eutropha* and recombinant *Escherichia coli*. The authors fermented these bacteria for purpose of PHA production, and Py-GC-MS was used to analyse the accumulated polymer samples in a short time, without any complex pre-treatment steps. In the obtained chromatogram, the authors reported the observation of major peaks of 2-butenoic acid, 2-pentenoic acid, and hexadecenoic acid derived from the PHA compositions and minor peaks of the cell components. Moreover, monomer compositions of poly(3-hydroxybutyrate-*co*-3-hydroxyvalerate) (PHBV) in *R. eutropha* were also determined based on the peak areas of 2-butenoic acid and 2-pentenoic acid by Py-GC-MS, which are the corresponding species of 3-hydroxybutyrate (3HB) and 3-hydroxyvalerate (3HV) in PHBV. The correlation between the conventional GC and Py-GC-MS data is extremely well matched. This finding evinced that Py-GC-MS can be a promising tool to rapidly screen PHA-accumulating strains based on polymer concentrations, as well as monomer compositions [66].

Gumel et al. [72] utilized GC-MS-MS to characterize the effects of different carbon sources on the PHA composition. The PHA samples in the form of hydroxyalkanoic acid methyl esters were subjected to GC-MS-MS analysis, and the chiral polymer indicated a composition of even and odd carbon atom chains with monomer lengths ranging from C4 to C14, including C8 and C10 as the main monomers [72].

### 3.4. SEC and GPC

Size exclusion chromatography (SEC) is a chromatographic method that separates compounds in a solution based on their size and molecular weight. This method is commonly employed to separate complex macromolecular substances, such as polymers and proteins. Gel-filtration chromatography is the terminology utilised when an aqueous solution (mobile phase) is used to transport the sample through the column (stationary phase), as opposed to the method of gel permeation chromatography (GPC), which is used when an organic solvent is used as a mobile phase. The chromatography column is filled with tiny, porous beads formed of agarose, dextran, or polyacrylamide polymers. These beads’ pore sizes are used to measure the dimensions of macromolecule samples. Because of its ability to generate accurate molecular weight data, SEC is commonly used for polymer characterization, which is deem as an advantage in the polymer research field.

Theoretically, the smaller molecules are retained in the pores of the adsorbent (stationary phase) in the column. Some particles enter the pores as the solution travels through the column. In contrast, larger particles cannot penetrate as many pores and simply pass by the pores, thus leading to a faster elution time. As a result, larger molecules move faster through the column than the smaller ones. In other words, longer retention times are associated with smaller molecules.

## 4. Spectroscopic Methods

Light and other forms of electromagnetic-radiation-based measurements are commonly used for polymeric material characterization. Spectroscopic analytical methods are based on the quantity of radiation generated or absorbed by the molecular or atomic species of the sample of interest. For example, NMR, IR, and X-ray methods are based on the measurement of the emission, absorption, scattering, fluorescence, and diffraction of electromagnetic radiation.

### 4.1. Nuclear Magnetic Resonance

NMR spectroscopy has long been utilised in the study of food samples, such as moisture, fat, and oil analyses [98], as well as biological samples [99]. As a consequence of creative advancements in the instrumentation, magnetic resonance imaging (MRI) and solid-state NMR are marketed, and their use has been expanded to various new domains in the past few years. In this subsection, special focus is placed on the uses of NMR in the research on biodegradable polymeric materials.

NMR spectroscopy has proven to be a method of choice for analysing the chemical structure and composition of polymeric biomaterials [68,100]. Nikolic et al. characterized the structure and average molecular weight of PBSA by ^1^H-NMR (200 MHz). The copolyester composition was also determined from the relative strengths of the proton signals formed by the succinate and adipate repeating units [101].

Moreover, Montaudo et al. synthesized and characterized a series of aliphatic copolyesters with number-average molecular weights ranging from 33,000 to 85,000. The co-polyesters consisted of 1,4-butanediol units paired with succinic, adipic, and sebacic acid units. Their compositions were calculated by integrating the ^1^H-NMR signals attributed to butylene succinate (2.628 ppm), butylene sebacate (2.294 ppm), and butylene adipate (2.332 ppm). The obtained composition values were in agreement with the feed ratio used during synthesis [102].

Although NMR provides accurate and precise compositional data for biodegradable polymers, NMR is considered as unsuitable for routine analyses, because it requires a relatively large sample size (a few tens of milligrams) as compared to THM-GC. Moreover, the NMR characterization of highly crystalline polymers, such as PHB, PBS, and chitin/chitosan, is cumbersome due to their low solubility in solvents. In this regard, THM-GC is deemed as the more practical and accurate method.

### 4.2. FTIR

The vibration of molecules in polymers is studied using Fourier-transform infrared (FTIR) spectroscopy. FTIR spectroscopy makes use of the way in which infrared light alters the dipole moments of molecules, which correspond to a certain vibrational energy. Because each functional group is formed of different atoms and bond strengths, the vibrations are unique to each functional group, such as O-H and C-H stretches, which emerge at approximately 3200 cm^−1^ and 2900 cm^−1^, respectively. Because the collection of the vibrational energy spectrum for all of a molecule’s functional groups is unique, these absorption bands may be utilized for the purpose of identification via library databases. Moreover, each functional group has its own unique vibrational energy, which may be utilized to identify a molecule when all the functional groups are combined. FTIR provides various advantages, including not only an increased light throughput but also its enhanced signal-to-noise ratio, simultaneous measurement of all wavelengths, speed of analysis, and sensitivity.

Jangong et al. reported the influence of extra polypropylene (PP) as a reinforcement in bioplastics containing starch/chitosan on the strong chemical bonding properties of functional groups in polymeric materials, as measured by the FTIR spectrometer model IRPrestige-21 (Shimadzu Corp.). The presence of O-H hydrogen bonds, C-H alkanes, C=C alkenes, and C-O alcohol compounds in the bioplastic was clearly observed in the acquired spectra. The FTIR measurements demonstrated that the hydroxyl groups have absorption band at 3432 cm^−1^. The spectra also portrayed C-H bonds at 2924 cm^−1^, C=C bonds at 1641 cm^−1^, and C-O bond formation at 1041 cm^−1^. For the polymeric materials, the C=C and C-H bonds showed a strong association with mechanical strain and bioplastic degradation rates. The authors also reported that the C-O bonds decreased with an increasing starch concentration. In this study, the authors highlighted that the presence of strong bonds with constituent materials improves the absorption intensity of C=C and C-H bonds, which enhances the tensile strength and biodegradation rate [69].

FTIR is also used to detect compounds in biodegraded materials by analysing the chemical bonds. For instance, FTIR tests revealed that fragments of biodegraded PLA and PHAs did not include any harmful by-products, such as cadmium, plumbum, phthalates, or dichlorodiphenyltrichloroethane (DDT) [59]. In separate studies, FTIR was used to screen and quantify intracellular PHA [103,104].

### 4.3. X-ray Diffraction

X-rays are a form of short-wavelength electromagnetic radiation created when decelerated high-energy electrons move into atoms’ inner orbitals. X-ray diffraction (XRD) examination is used by polymer scientists to analyse solid-state structural characteristics, such as crystallinity and amorphous polymeric and composite materials [105].

Jangong et al. fabricated starch/chitosan reinforced with PP to produce highly biodegradable polymers in various ratios of 35:65, 50:50, and 65:35. The obtained polymers were subjected to XRD analysis to elucidate their degree of crystallinity. The Shimadzu 7000 X-ray diffractometer was utilized with CuKα radiation (λ = 1.5405 Å) and data recorded within 15° ≤ 2θ ≤ 60°, operating at 30 kV and 10 mA. The obtained spectra portray the intensity of the XRD peaks (a.u) versus the 2θ (degree) of CuKα. Overall, the acquired results revealed that the biopolymer had an amorphous crystalline structure, with the major wide peaks located between 18° and 30° [69].

### 4.4. X- ray Fluorescence

For the analysis of trace elements in polymers, X-ray fluorescence (XRF) is the ideal method. Inorganic additives, such as plasticizers, lubricants, stabilizing agents, neutralizers, antioxidants, and pigments, as well as catalytic agents, can be determined through this non-destructive nature of XRF. Furthermore, the sample preparation is seldom required, thus rendering this method applicable to a wide range of samples and multipurpose analytical techniques.

Shruti et al. characterized 33 randomly collected single-use plastics (SUPs) in Mexico, which are typically claimed to be oxo-biodegradable and biodegradable, and elucidated their polymer compositions and metal contents. The metal concentrations were identified and estimated using an energy-dispersive XRF model EDXRF, Niton FXL 950, with a 50 kV X-ray tube constructed from argentum, and compared them to the reference standards for packaging materials from various countries. Among the 25 various metals detected, most of the SUPs exceeded the standard international values, and the highest concentrations of Cu, Cr, Mo, Zn, Fe, and Pb were 1898 mg/kg, 1586 mg/kg, 95 mg/kg, 1492 mg/kg, 1900 mg/kg, and 7528 mg/kg. Based on these discoveries, the authors highly recommended the implementation of the strict quality assurance of plastic production and manufacturing improvements of the SUPs to ensure their claimable criteria as commercial biodegradable plastics. SUPs typically consist of 85% high-density PE and 15% low-density PE, as well as inorganic compounds. These chemicals compounds may slowly leak into the ecosystem in response to light or heat and lead to undesirable issues such as health and ecological implications. The authors also integrated the elemental composition data obtained by means of XRF together with SEM analysis for the polymer surface morphology characterization [70].

## 5. Respirometric Methods

Respirometry can be defined as a set of techniques for estimating the indirect measurement of the respiration metabolism rates of microorganisms, cells, tissues, plants, vertebrates, or invertebrates. This method can be manipulated to access information related to the biodegradation mechanisms of polymeric samples due to the activity of microorganisms.

### Biogas Evolution

In polymer degradation, any emitted gases, such as carbon dioxide (CO_2_), released from the decaying samples submerged in water or any solution in the sample jars are collected and measured by means of respirometric techniques [106,107]. The CO_2_ biogas is measured using a sensor with a computerized controlling program [59]. In another study, by measuring the biochemical oxygen demand (BOD) biodegradation, it was possible to correlate the proportion of biodegradation of the samples with their time spent in water. Data on BOD biodegradation evinced that the materials were broken down by microbial metabolism activity. The typical procedure involved placing water or another solution together with samples of interest in a fermenter and measuring the amount of consumed oxygen (O_2_) gas. Normally, as a control experiment, another set of fermenters with no sample are prepared. The net amount of O_2_ gas consumed by the microbes (in mL units) was estimated by the difference between the samples and the control. This theoretical amount of consumed O_2_ is the total O_2_ that is utilized during complete sample degradation [60].

## 6. Meta-Analysis

Conceptual meta-analysis is a statistical technique that can be conducted by merging multiple scientific research data assessing the biodegradation rates of polymeric materials, with each individual study providing data with a tolerable degree of error. According to Dilkes-Hoffman et al., the meta-analysis of PHA biodegradability gave a result of 0.04–0.09 mg/day/cm, which equates to a period of 1.5–3.5 years required for a PHA water bottle to completely breakdown in coastal ecosystems. The authors conceptualized the biodegradation mechanisms using three steps, viz., (1) biofilm growth and surface degradation, (2) enzymatic-catalysed and hydrolytic depolymerization, and (3) the microbial absorption of micro-compounds [71]. The advantage of this method is that it enables the data projection to quantitatively predict the duration time required for a certain biopolymer to degrade. Since some biopolymers require a longer degradation timescale of a few years, by collecting data such as the mass reduction and size of the polymers in the initial few months of the biodegradation test, these obtained data can be manipulated by means of meta-analysis, and the prediction of the total duration of time required for a certain polymeric material to degrade can be estimated.

In polymer testing, the goal of meta-analysis is to identify all the variations in the data matrix, analysed using data consisting of multiple variables derived from a number of samples. This method can identify the relationships between the samples and variables in a given dataset and convert them into new variables. The development and advancement of various forms of software used in meta-analysis have led to new possibilities for statistical technique improvement. Multivariate decomposition methods, multivariate regression methods, pattern recognition methods, hierarchical cluster analysis, chemometric resolution methods, and many more are among the examples of statistical analytical methods. The method selection is based on the estimated complexity of the data [108].

## 7. Thermal Methods

Thermal methods, also known as thermogravimetric analysis, are a group of techniques through which specific properties of polymeric materials are measured with the change in temperature. Several methods are routinely used in polymeric material analyses, including differential scanning calorimetry (DSC), differential thermal analysis (DTA), and thermogravimetric analysis (TGA). Thermal analysis is commonly used for polymer fingerprinting and characterization. Thermal stability is a crucial characteristic for investigation, since it represents the highest temperature that polymers can withstand prior to undergoing thermal degradation.

### 7.1. Differential Scanning Calorimetry

Differential scanning calorimetry (DSC) is the process of applying a linear heating or cooling signal to a material, followed by measurements of the temperature and enthalpy (energy), which is correlated with a variety of thermal processes, such as melting, crystallization, glass transitions, and decomposition reactions [109]. Information on the heat capacity (*C_p_*), glass transition temperature (*T_g_*), crystallization (*T_c_*), thermal degradation temperature (*T_d_*), and melting point (*T_m_*) of a polymer sample can be obtained. The advantages of this method are its applicability to wide temperature ranges as low as −180 °C and as high as 2400 °C, its rapidity, and the fact that it is user-friendly. Normally, a standard sample with 100% crystallinity is used to calculate the enthalpy.

Baidurah et al. employed DSC to determine the crystallinity of a PBSA film. THM-GC analysis was also performed to monitor changes in the copolymer content. The authors created a PBSA film with a low degree of crystallinity to investigate the influence of crystallization on the degradation rate. The results showed that the biodegradation rate was greatly accelerated in the film with a lower crystallinity and high butylene adipate (BA) moieties. As a result, they proposed that the BA-rich moieties were preferentially biodegraded during the soil burial test, resulting in a decreasing BA content with the elapsing degradation time [48].

### 7.2. Differential Thermal Analysis

The application of data derived from differential thermal analysis (DTA) is almost the same as that of DSC. In this technique, the temperature difference between the sample and a reference material is monitored against the time or temperature, whilst the temperature of the sample is programmed under a specified atmosphere condition. DTA is often integrated with thermogravimetric analysis (TGA), and its further elaboration is addressed in the following subsection.

### 7.3. Thermogravimetric Analysis

To elucidate weight changes as a function of temperature and time in polymeric materials, thermogravimetric analysis (TGA) is often employed. Decomposition and oxidation reactions, as well as physical processes, such as sublimation, evaporation, and desorption, can cause weight changes in polymeric materials. Hyphenated or integrated TGA combined with other instruments, such as TGA-DTA, TGA-FTIR, TGA-MS, and TGA-GC-MS, can enhance the data acquisition. Gumel et al. successfully biosynthesized medium-chain-length PHA accumulated intracellularly by *Pseudomonas putida* Bet001 isolated from palm oil mill sludge. Thermal analyses indicated the PHA was semi-crystalline and portrayed an excellent thermal stability, with a *T_d_* of 264.6 to 318.8 °C, *T_m_* of 43 °C, *T_g_* of −1.0 °C, and a melting enthalpy of fusion (Δ*H_f_*) of 100.9 J/g. The TGA also indicated an enhanced polymer stability in correlation with the increasing fatty acid chain length, ranging—for example—from C8:0 to C18:1, with the thermal degradation temperature between 264.6–318.8 °C. This observation can be explained by an increase in the proportions of longer monomers, such as 3-hydroxydodecanoate (3HDD) and 3-hydroxytetradecanoate (3HTD), integrated into the PHA polymer backbone through the metabolism of the longer fatty acid substrates [72].

## 8. A Comparison of the Aforementioned Methods

The key factor in determining the method selection is mainly based on the availability of instruments in certain institutions. Evidently, many researchers have employed various methods to study different principles that can measure the distinct properties of any biodegradable polymer. For example, DSC elucidates the latent heat of the melting of crystalline regions; XRD determines the solid-state structural properties, such as the degree of the crystallinity domains; NMR monitors the chemical structure and compositional data; and FTIR analyses the variation in the strengths of chemical bonds, and the findings can be interpreted in terms of specific functional groups. Consequently, the combination of data obtained using these various methodologies enables one to achieve a clearer picture of the polymer structure at both the micro- and macroscopic levels.

A comparison of the P(HB-*co*-HV) and PBSA degradation mechanisms in laboratory-scale composting settings assessed by respirometric methods, the mass loss, DSC, GPC, FTIR, and SEM, which was conducted by Salomez et al. (2019), indicated that both polymers deteriorated drastically in less than 80 days. Due to its high molecular weight and degree of crystallinity, P(HB-*co*-HV) degraded more rapidly than PBSA, which indicated that two biodegradation mechanisms were involved. Based on these two biodegradation mechanisms, the disparity in the biodegradation rate between P(HB-*co*-HV) and PBSA is primarily due to major differences in the crystal morphology and spatial arrangement of the two polymers. Concerning the significance of the various indicators tested, the mass loss was touted as the most pertinent indication for assessing the disappearance of the polymeric material, especially when compared with the respirometric method [74].

Nevertheless, in more recent findings by Shruti et al., the authors implied that to successfully evaluate and comprehend the single-use plastic properties, multiple analytical methods, including techniques such as FTIR, XRF, TGA, and DTA, must be conducted [70].

The THM-GC, NMR, and DSC methods are unquestionably the most efficient in terms of convenience, ease of the sample preparation, and precision. The NMR and DSC methods are extremely adaptable and can be used to measure a broad range of biodegradable polymers. As the NMR measurements are non-destructive, many measurements can be obtained from the same polymeric material during the biodegradation test.

## 9. Conclusions

This review summarized the latest information and advancements regarding various methods suitable for the study of biodegraded polymers. Knowledge related to the advantages, limitations, techniques, and precautions of each method was discussed. The researcher has a variety of methodologies, and the selection is inevitably influenced by the facilities available, as well as the experiment’s aims. The researcher should be aware of any limitations of the selected method. In any situation, it may be prudent to select a minimum of two methods, so that the findings of one method may be confirmed and cross-checked with those of other methods. With the data obtained from various analytical methods, polymer scientists are able to tailor the degradation rates of polymeric materials based on their final applications.

## 10. Future Outlook

It is forecast that the consumption and production patterns of biopolymers will expand in the coming years based on the current scenario of the market size, share, growth, demand, and trends. However, the high cost of biopolymer production remains one of the greatest barriers to its ability to compete with petroleum-derived plastics. Continuous technological advancement and invention may contribute to the minimization of the cost barrier, rendering it a viable option for industry sectors. With the analytical technology advancement, various analytical methods have become available for biopolymer characterization, including that of severely degraded polymeric materials.

## Figures and Tables

**Figure 1 polymers-14-04928-f001:**
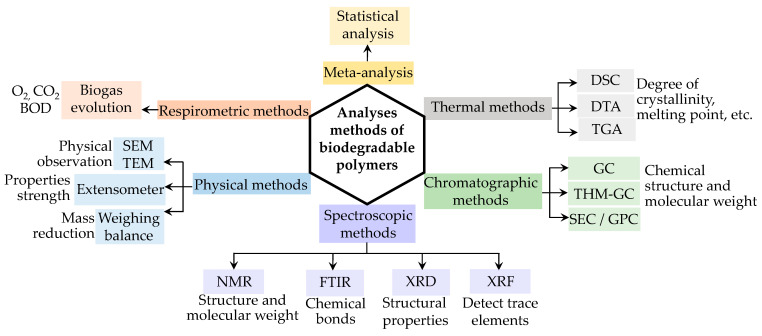
Analytical methods of biodegradable polymers.

**Table 1 polymers-14-04928-t001:** Classification of typical commercially available biodegradable polymers based on their origins, together with the trade names and manufacturers [9,10].

Origin	Polymer Name	Trade Name	Manufacturer
Produced by bacteria	Polyhydroxyalkanoates (PHAs),Polyhydroxy butyrate (PHB) 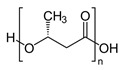	Biopol^TM^	Monsanto Company (St. Louis, MO, USA)
Polyhydroxybutyrate-*co*-hydroxyvalerate [P(HB-*co*-HV)] 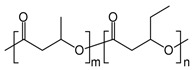	Imperial Chemical Industries (London, UK)
Gellan gum	Kelcogel^R^	Kelco Biopolymers (Atlanta, GA, USA)
Curdlan	Pureglucan^R^	Takeda Chemical Industries (Osaka, Japan)Nuture, Inc. (New York, NY, USA)
Produced from natural products and their derivatives	AmyloseAmylopectineStarch	Mater-Bi^R^,Vegemat^R^	Novamont (Novara, Italy)Vivadur (Bazet, French)
Chitin	-	Eastman Chemical Company (Kingsport, TN, USA)Primester (Kingsport, TN, USA)Celanese Ltd. (Irving, TX, USA)
Cellulose acetate (CA)	ChitoPure^R^	USA Biopolymer Engineering (Saint Paul, MN, USA)
Produced via chemical synthesis	Polyglycolic acid (PGA) 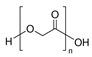	Biomax^R^	Dupont (Wilmington, DE, USA)
Polylactic acid (PLA) 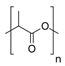	NatureWorks^TM^	Cargill Dow (Plymouth, MN, USA)Mitsui Toatsu Chemical (Tokyo, Japan)
Polycaprolactone (PCL) 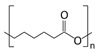	Capa^R^	Solvay Group (Brussels, Belgium)
Poly(butylene succinate-*co*-butylene adipate) (PBSA)	Bionolle	Showa Highpolymer (Tokyo, Japan)
Poly (butylene succinate) (PBS) 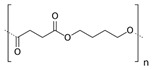

## Data Availability

Not applicable.

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
