# Peer review of "Methods of Analyses for Biodegradable Polymers: A Review"

_polymers, 2022, doi:10.3390/polym14224928_

Round 1

Reviewer 1 Report

The reviewed manuscript titled ‘Methods of Analyses for Biodegradable Polymers: A review’ was submitted as a review type manuscript. The title of the manuscript corresponds to its content. In Introduction, Author provides an essential information about biodegradable polymers: characterizes types of biopolymers, methods of their synthesis, ways of biodegradation and indicates purposes and methods of their analysis. In the following sections, the more detailed characterization and application of different analysis methods for biodegradable polymers characterization is given. The concept of the work is good and interesting. However, it is necessary to make some corrections and clarifications before publishing the manuscript.

1.     First of all, English needs serious correction. In places the article is written in a very clumsy manner, which forces the reader to guess the author's intentions.

2.     Line 49: what does it mean “green polymetric polymers’?

3.     Line 177: Author wrote ‘The large size of porous surface area can be interpreted as the degree of biodegradation.’ This statement does not make sense. I wonder, how can a porous surface be related to the degree of biodegradation?

4.     Next, the same paragraph, sentence in lines 181-183: this sentence does not contribute anything to the work, it is not known what kind of samples are meant.

5.     Line 193: please give more information about the biopolymer used in the cited paper.

6.     Table 2, row 12: DSC it is not the method for monitoring the reduction of molecular weight of polymers! Please correct this statement.

7.     Table 2, row 3: please explain the abbr. THM. It is not an obvious abbreviation that is used commonly.

8.     Table 2, row 6: the sentence ‘To isolate efficient PHAs producing bacteria derived from sludge soil.’ is misleading. It should be corrected since neither FTIR or NMR are isolation methods.

9.     Line 237: Author wrote ‘So far, this technique is widely used to elucidate the composition of numerous volatile biodegradable polymers, such as PHB’. I couldn’t agree with Author. Polymers/biopolymers are not volatile compounds that can be analyzed directly by GC method. Please correct this sentence.

Moreover, please explain why methylation is preferably performed before GC analysis.

10.  I wonder, why Author wrote ‘copolymer’ word in different manners thorough the text. The spelling should be unified.

11.  Line 317: please give more conclusions emerge from cited paper 82.

12.   Paragraph 3.3: please show some examples of using GC-MS for biopolymers characterization.

13.  The sentence starting from line 351: ‘Furthermore, a decrease in molecular weight leads to a reduction in melting temperature, which may also be determined via SEC and cross-check with differential scanning calorimetry’ – it is nonsense. It is impossible to determine melting temperature by DSC! Moreover, Author cite in this place ref. 44 and 97. These works are not properly cited here: in ref 44 and 97 no results from the DSC or at least the melting temperatures are discussed.

14.  Line 371: in NMR the term signal is preferable than peak.

15.  Line 393: in FTIR the term absorption band is preferable than peak.

16.  Line 403: please explain what do you mean writing ‘the strong chemical bonding properties’.

17.  Lines 409-411: Author cite the results form ref. 69 and wrote that ‘however the C-O bonds decrease with increasing starch concentration’. This statement is not consistent. It should be rewritten.

18.  Line 503: replace ‘equipment’ by ‘methods’.

19.  Line 518: please rewrite the sentence, since the DSC in not used for monitor the molecular weight of polymers. Moreover, the ref. 97 is not properly cited in this place (look p.13).

Reviewer 2 Report

The submitted Review can be accepted after rechecking and considered the following items:

There many typos should be corrected, In line 43, can decompose should be corrected to can be decomposed.

I suggest for the authors to provide a section about the different applications of biodegradable polymers.

Future outlook should be written and added after conclusion part.

I suggest for the authors to fabricate subtitles about the advantages and disadvantages of the utilized biodegradable polymers in different applications, especiall

The review has been submitted to polymers journal, thus, the authors should highlight and draw the chemical structure of some of the biodegradable polymers.

Round 2

Reviewer 1 Report

Authors revised the manuscript according my remarks. In my opinion, the manuscript can be published in present form.

I have one more point: writing in my review 'what does it mean green polymetric materials' I didn't think about the word 'green' but 'polymetric'. Please correct it, in line 48, 92, 130

Besides, I owe the authors an explanation concerning remark no.13: I made a mistake in sentence 'It is impossible to determine melting temperature by DSC' it should be of course SEC.